# Dynamics of a Protein Interaction Network Associated to the Aggregation of polyQ-Expanded Ataxin-1

**DOI:** 10.3390/genes11101129

**Published:** 2020-09-25

**Authors:** Aimilia-Christina Vagiona, Miguel A. Andrade-Navarro, Fotis Psomopoulos, Spyros Petrakis

**Affiliations:** 1Department of Biology, Aristotle University of Thessaloniki, 54124 Thessaloniki, Greece; emilyvagiona@gmail.com; 2Faculty of Biology, Johannes Gutenberg University, Biozentrum I, Hans-Dieter-Hüsch-Weg 15, 55128 Mainz, Germany; andrade@uni-mainz.de; 3Institute of Applied Biosciences/Centre for Research and Technology Hellas, 57001 Thessaloniki, Greece; fpsom@certh.gr; 4Department of Molecular Medicine and Surgery, Karolinska Institutet, 17177 Stockholm, Sweden

**Keywords:** ataxin-1, polyQ, pathway, network, drugs, blood-brain-barrier

## Abstract

Background: Several experimental models of polyglutamine (polyQ) diseases have been previously developed that are useful for studying disease progression in the primarily affected central nervous system. However, there is a missing link between cellular and animal models that would indicate the molecular defects occurring in neurons and are responsible for the disease phenotype in vivo. Methods: Here, we used a computational approach to identify dysregulated pathways shared by an in vitro and an in vivo model of ATXN1(Q82) protein aggregation, the mutant protein that causes the neurodegenerative polyQ disease spinocerebellar ataxia type-1 (SCA1). Results: A set of common dysregulated pathways were identified, which were utilized to construct cerebellum-specific protein-protein interaction (PPI) networks at various time-points of protein aggregation. Analysis of a SCA1 network indicated important nodes which regulate its function and might represent potential pharmacological targets. Furthermore, a set of drugs interacting with these nodes and predicted to enter the blood–brain barrier (BBB) was identified. Conclusions: Our study points to molecular mechanisms of SCA1 linked from both cellular and animal models and suggests drugs that could be tested to determine whether they affect the aggregation of pathogenic ATXN1 and SCA1 disease progression.

## 1. Introduction

The polyglutamine (polyQ) disease spinocerebellar ataxia type-1 (SCA1) is a lethal, progressive, autosomal dominant neurodegenerative disorder caused by a CAG trinucleotide expansion in the ataxin-1 (ATXN1) gene [1]. This mutation produces a longer polyQ tract in the pathogenic protein which gradually misfolds into an abnormal conformation and forms protein inclusions within the nuclei of neurons [2]. Cerebellar neurons that coordinate movement are particularly sensitive to ATXN1 aggregation; their gradual dysfunction and loss are responsible for the characteristic symptoms of SCA1, including loss of coordination and ataxia [3,4].

Formation of polyQ inclusions is the main feature of SCA1 pathology; however, other factors also affect the progression of the disease. Several studies have shown that ATXN1 interacts with transcription regulators, RNA splicing factors and other nuclear receptors [5,6,7,8], suggesting that alterations of these interactions by the polyQ expansion in the mutant protein might drive cerebellar pathology [9]. To develop treatments for this disease, it might beneficial to employ approaches from network medicine, which studies disease in terms of the pathological effects of altered protein interaction networks [10,11,12]. Neurodegenerative diseases characterized by protein aggregates, which spread in the nervous system [13], have been studied from this point of view [14,15].

In order to gain insights into the pathogenesis of SCA1, various experimental models of the disease have been previously generated, including induced pluripotent stem cells [16] and transgenic mice expressing human ATXN1, with an expanded polyQ tract [17,18]. SCA1 B05 mice are widely used for modeling this disease in vivo, since they develop Purkinje cell degeneration, indicating that a mouse model can be established simply by introducing CAG repeat expansions in a wild-type protein. Purkinje cells gradually accumulate nuclear inclusions which increase with age; approximately 90% of these neurons contain polyQ inclusions by week 12. In parallel, mice at increasing ages, (week 5, 12 and 28) show symptoms corresponding to mild, moderate and severe ataxia, respectively [17]. Recent studies showed that ATXN1 expression levels are regulated by miR760 binding to a conserved region in its 5’ untranslated region [19], while the ATXN1-Capicua (CIC) protein complex is the main driver of pathology in the cerebellum through a gain-of-function mechanism [9]. Furthermore, analysis of the cerebellar transcriptome of SCA1 B05 mice indicated gene networks whose expression profiles correlate with disease progression in the cerebellum [20]. These networks were also dysregulated in a different SCA1 in vivo model [18] and differ considerably to other regions of the brain, including the medulla [21].

We previously developed an inducible cellular model of ATXN1(Q82) protein aggregation in human mesenchymal stem cells (Tet-On YFP-ATXN1(Q82) MSCs), in which the pathogenic protein gradually forms insoluble intranuclear inclusions. These inclusions cause oxidative and nucleolar stress, affect the assembly of the ribosome and eventually lead to cell necrosis. Furthermore, a number of transcriptional changes were identified which correlate with the gradual aggregation of polyQ-expanded ATXN1 in human SCA1 cerebellum, as well. These include dysregulation of the protein synthesis machinery and pathways involved in focal adhesion or oxidative phosphorylation [20]. This inducible model offers the possibility to identify specific molecular changes in vitro which may contribute to the selective neuropathological phenotype in vivo.

Here we attempted to identify common molecular changes in Tet-On YFP-ATXN1(Q82) MSCs and SCA1 B05 mice. Proteins participating in dysregulated pathways at different time–-points of polyQ aggregation were used for the construction of perturbed cerebellum-specific protein-protein interaction (PPI) networks. Our analysis indicates important nodes of a SCA1 PPI network affected by the gradual aggregation of pathogenic ATXN1. Pharmacological targeting of these proteins may modify polyQ aggregation and SCA1 disease progression.

## 2. Materials and Methods

### 2.1. Datasets of SCA1 B05 Transgenic Mice and Tet-On YFP-ATXN1(Q82) MSCs

RNA-seq datasets from the cerebellum of three age groups (*n* = 3 animals per group) of SCA1 B05 transgenic mice (week 5, week 12 and week 28) and age-matched control FVB mice were retrieved from the literature [21]. The GEO accession number of these RNA-seq data is GSE75778. The DIOPT (DRSC Integrative Ortholog Prediction Tool) tool, which integrates ortholog predictions from 11 commonly used orthology tools [22], was used to map human orthologs of murine genes. Only human genes with a high rank score were selected.

RNA-seq datasets from Tet-On YFP-ATXN1(Q82) MSCs at three different time-points of polyQ-expanded protein aggregation (Day 2, Day 5 and D10) and a control time-point (Day 0) (*n* = 3 samples per time point) were selected. These data are publicly available [20]. The final list contained only genes that are specifically dysregulated by the expression of pathogenic ATXN1.

### 2.2. Differential Gene Expression Analysis

Gene expression levels were measured in “fragments per kilobase of exon model per million mapped reads” (FPKM) values [23]. A mean FPKM value from the experimental triplicates was calculated for each gene. For differential gene expression analysis, a fold change (FC) for each gene was calculated using the following equation and FC data were log_2_ normalized.
FC = [FPKM g (SCA1)]/[FPKM g (control)]

Genes from SCA1 B05 transgenic mice (week 5–week 28) were compared to FVB control mice at the same age; genes from Tet-On YFP-ATXN1(Q82) MSCs at Day 2–Day 10 were compared to cells at Day 0. A *t*-test was applied to compare FPKM levels between a group and its respective control. Genes with |log_2_FC| > 0.5 and *p*-value < 0.05 were considered as DEGs (differentially expressed genes) and used for further analysis. Differential expression analysis was performed using R version 3.6.1 programming software (RStudio Team 2016).

### 2.3. Pathway Enrichment Analysis

DEGs were used for pathway enrichment analysis in the online tool Enrichr (version 2.1) [24]. Dysregulated pathways in SCA1 B05 transgenic mice and Tet-On YFP-ATXN1(Q82) MSCs were identified using the Kyoto Encyclopedia of Genes and Genomes (KEGG) database and were ranked by *p*-value, as calculated by the EnrichR platform. KEGG pathways from mice and cells were compared and common pathways at each time-point with a *p*-value < 0.05 were selected.

### 2.4. Construction of PPI Networks

Protein components of the common dysregulated pathways were used for the construction of PPI networks using the STRING database [25] in Cytoscape (version 3.7.2) [26]. Only genes expressed in the nervous system (score of 4.8 using the relevant tissue filter) [27] and high confidence protein interactions (score of 0.950 or above) were utilized. Unconnected nodes were deleted. Statistical significance for the networks was measured using the Motif Discovery plug-in of Cytoscape (version 0.0.3). A z-score for 4-node motifs was calculated for each network after comparison with 1000 random networks. The topological properties of the networks were calculated using the Network Analyzer plugin of Cytoscape [28] and were visualized using the GraphPad Prism software.

### 2.5. Selection of Genes Analysis

The online tool Génie was used to select mouse genes associated to the biomedical research in protein aggregates in the brain [29]. Basically, this tool uses a user-defined query to PubMed to retrieve a non-specific set of records that is used to define a score for the words enriched in the set. In a second search, PubMed records are evaluated for their content in the enriched words and genes from a species linked to those records are ranked. Génie also allows gene selection via orthologs. Here, we used Génie with default thresholds, using the query “protein aggregation brain” to rank mouse genes, using human orthologs to extend the literature.

### 2.6. Network Analysis

NetworkAnalyzer was used to compute the topological centralities of PPI networks. Genes were extracted based on three criteria: (a) degree centrality (DC), (b) betweenness centrality (BC) and (c) closeness centrality (CC).

DC of each node, i, was defined as:D(i) = Σjm(i,j)
where m(i,j) = 1 if there is a link from node i to node j.

BC referring to the frequency of node i appearing at nodes j and k was calculated by the equation:B(i) = Σa, bgjik/gjk
where i ≠ j ≠ k, gjk is the number of the shortest pathways between nodes j and k, gjik is the number of the shortest pathways containing i.

CC of a node i in a graph was calculated by the equation:C(i) = Σjd(i,j)
where i ≠ j, dij is the shortest pathway between nodes i and j.

Important nodes/proteins were identified based on topological networks analysis.

### 2.7. Drug-Protein Interaction Network

A drug-protein interaction network was constructed for selected proteins (coefficient score of 0.5) using the Cytoscape plugin CyTargetLinker [30]. Drug-protein interactions were retrieved from the DrugBank database [31]. The online tool Blood Brain Barrier Predictor [32] was utilized to assess whether selected drugs can penetrate the blood-brain barrier (BBB).

## 3. Results

### 3.1. Identification of Dysregulated Pathways in Tet-On YFP-ATXN1(Q82) MSCs and SCA1 B05 Transgenic Mice

First, we attempted to identify dysregulated pathways in two different experimental models of polyQ-expanded human ATXN1, which show a consistent dysregulation in gene expression due to the aggregation of the mutant protein. The computational workflow used here is summarized in Figure 1. Gene datasets were selected from an in vitro cell model and an in vivo mouse model of protein aggregation, namely, Tet-On YFP-ATXN1(Q82) MSCs and SCA1 B05 transgenic mice, respectively. Several of the selected DEGs, at least in the in vitro cell model, were independently validated by quantitative MS [22]. Importantly, both models expressed full length human ATXN1 harboring the same pathogenic polyglutamine length. In order to compare the two datasets, murine genes from the SCA1 B05 transgenic mice were first converted into their human orthologs. Then, the change in their expression levels was calculated at three different time-points of protein aggregation [week 5 (W5)—early, week 12 (W12)—middle and week 28 (W28)—late] compared to the respective age-matched control group (FVB mice). Genes with a mean |log_2_FC| > 0.5 and < 0.05, indicating a consistent expression among triplicates, were considered as DEGs and were used for further analysis. Similarly, using the same criteria, DEGs were selected from Tet-On YFP-ATXN1(Q82) MSCs at three different time-points of protein aggregation [Day 2 (D2)—early, Day 5 (D5)—middle and Day 10 (D10)—late] (Appendix A). These cells gradually accumulate insoluble polyQ-inclusions compared to control [Day 0 (D0)] Tet-On TFP-ATXN1(Q82) MSCs. The majority of selected DEGs in both Tet-On YFP-ATXN1(Q82) MSCs and SCA1 B05 transgenic mice were downregulated, as previously described [20,21]. The two datasets shared 143 genes which showed a significant agreement in the direction of fold change with 38 genes upregulated in at least one time-point in both models (*p*-value 0.011): these 38 upregulated genes were functionally enriched for genes involved in extracellular matrix organization (Benjamini corrected *p*-value 1.7 × 10^−6^), but not the 33 genes with downregulation in both models or the remaining ones. Furthermore, these 143 common genes are enriched in binding sites for the ETV4 transcription factor (Appendix A) whose activity is regulated by CIC, the main driver of pathogenesis in SCA1 [33].

In order to identify dysfunctional pathways associated with gradual protein aggregation, DEGs in each dataset and time-point were used for enrichment analysis using the KEGG database. Only pathways with a *p*-value < 0.05 were considered as significantly dysregulated at each time-point (Figure 1). Using the DEGs at D2 cells (*n* = 687) and W5 mice (*n* = 357), this analysis identified 28 and 18 pathways, respectively, that were dysregulated at an early time-point of polyQ protein aggregation in the two experimental models. Similarly, DEGs at D5 cells (*n* = 789) and mice at W12 (*n* = 1204) were categorized in 36 and 58 pathways, respectively. Finally, DEGs (*n* = 801 at D10 cells and *n* = 1063 at week 28 mice) indicated 32 pathways in each experimental model that were associated with a late stage of polyQ-expanded ATXN1 protein aggregation (Appendix A).

We hypothesized that the gradual aggregation of ATXN1(Q82) into insoluble inclusions in MSCs may have similar molecular characteristics to progressive ataxia observed in B05 mice with aging. Therefore, we matched the selected time-points from cells and mice (D2 MSCs to W5 mice, D5 MSCs to W12 mice and D10 MSCs to W28 mice) and asked which dysregulated pathways were common in both protein aggregation models at each time-point. Three pathways, protein digestion and absorption, ECM-receptor interaction and PI3K-Akt signaling pathway were identified at an early stage of protein aggregation. Interestingly, these pathways were dysregulated at all further time-points in both models. At a middle stage, five more pathways were identified, including ribosome biogenesis, Alzheimer’s and Parkinson’s disease. Two of them, Rap1 signaling pathway and focal adhesion were dysregulated also at a later time-point. Furthermore, regulation of actin cytoskeleton and AGE-RAGE signaling pathway specifically featured the later stage, when polyQ-expanded ATXN1 forms terminal inclusions (Table 1)

### 3.2. Perturbed PPI Networks in SCA1 Models

Following previous strategies that studied the mechanisms of neurodegenerative disease from a network medicine perspective [14], we generated PPI networks associated with the gradual aggregation of pathogenic ATXN1 using as input the proteins that participate in the common dysregulated pathways of the SCA1 models. Three perturbed PPI networks were generated at each time-point (D2 cells/W5 mice: z-score = −0.981, D5 cells/W12 mice: z-score = −0.990 and D10 cells/W28 mice: z-score = −0.976), which included only high-confidence PPIs of proteins produced in the nervous system (Appendix A). The largest network was observed at a middle stage of protein aggregation. The protein nodes of these networks and the pathways in which they participate are shown in (Appendix A). These networks were significantly enriched in genes related to protein aggregates in the brain, particularly the middle stage network (see Methods for details) (Appendix A). We note that aggregation-related genes in the early network were also present in the middle and late networks, but aggregation-related genes present in the middle network were not in the late network.

Then, PPIs of these networks were combined into a large network, which is perturbed by the gradual aggregation of pathogenic ATXN1 (Figure 2A). The SCA1 PPI network (z-score = −0.977) contains discrete clusters and subnetworks associated with various stages of protein aggregation, perturbed either at all time-points (*n* = 50, yellow color), or specifically at a middle or late time-point (*n* = 87 green or *n* = 38 magenta color, respectively) (Figure 2B and Appendix A). Specifically, the yellow cluster of ATP1 proteins is involved in sodium ion transport and is perturbed at all time-points of protein aggregation (Appendix A). The green cluster participates in ATP synthesis as most of these proteins are components of the mitochondrial respiratory chain (Appendix A). Finally, ribosomal proteins and proteins participating in G-protein signaling pathway form highly interconnected modules within the SCA1 PPI network (Appendix A, respectively).

### 3.3. Analysis of SCA1 PPI Networks

Next, we analyzed the connectivity and functionality of the SCA1 PPI network by calculating the degree (DC), betweenness (BC) and closeness (CC) centralities of its components at all time-points (early, middle and late stage of protein aggregation). We identified 11 proteins (early: PPP2CA, TP53, MTOR and PIK3R, middle: RPS6, RPL15 and RPS3, late: CDC42, RHOA, PIK3R1 and CTNNB1) with the highest DC values, namely, the number of connections of each node. These proteins were forming important hubs within the individual networks (Appendix A).

However, the importance of a node in a biological network does not depend only on its number of neighbors [34] but it may increase as a node participates in communication paths and controls the flow of information (BC) [35] or it has a central role in the network being closer to all the other nodes (CC). Thus, we identified important nodes with the highest BC values (early: GNB1, GNB5 and MTOR, middle: CD44, PIK3R2 and YWHAH, late: CDC42, RHOA and GNB1) (Appendix A). On the other hand, ATP1 protein subunits (ATP1A1, ATP1A2, ATP1A3, ATP1B1, ATP1B2 and ATP1B3), which formed a discrete cluster within the network, had the highest CC values at all three stages of protein aggregation (Appendix A). The numerical values of DC, BC and CC for all nodes of the SCA1 PPI network are show in Appendix A.

Centrality changes may indicate whether a node gains or loses its importance within a network [36]. Therefore, we attempted to link time-dependent changes in the centralities of the highly ranked nodes with the dynamic changes of the SCA1 network, which trigger various cellular mechanisms as the disease progresses. In total, 21 proteins were selected based on their DC, BC and CC values. The changes in their centralities during polyQ protein aggregation are visualized in Figure 3. These heatmaps indicate that CDC42 and RHOA, which are involved in the regulation of the cell cycle and the formation of stress fibers [37], gradually increase their DC and BC (Figure 3A,B). In contrast, GNB1 and GNB5, which are involved in the transduction of various transmembrane signals in cells, including neurons [38], show a significant decrease in their BC and CC (Figure 3B,C). Furthermore, the CC of CD44, a cell-surface receptor involved in cell adhesion and regulating the functionality of dendritic spines [39] is dramatically decreased (Figure 3C). In contrast, ATP1 subunits do not show any obvious change in their centralities at all time-points (Figure 3).

### 3.4. Drug-Protein Interaction Network in SCA1

Several lines of evidence indicate that nodes of a network with increased DC and BC values represent potential drug targets [40,41]. In contrast, reduced centrality may characterize nodes that gradually lose their function and need to be stimulated. A number of SCA1 network components (GNB1, PPP2CA, MTOR, TP53, CDC42, RHOA and ATP1A1) were further prioritized as potential targets for pharmacological intervention. These proteins were considered important regulatory nodes of the SCA1 network, based on the gradual change of their centralities (Figure 3). GNB1, PPP2CA, MTOR and TP53 encode proteins with decreased BC and CC centralities, while DC, BC and CC centralities of CDC42 and RHOA increased over time. Additionally, ATP1A, a representative component of the discrete cluster of the network was also selected, based on its constantly high CC value. A drug-protein interaction network was constructed, depicting the interactions between 7 target-proteins and 32 drugs which are which are either FDA-approved or under evaluation in clinical trials (Appendix A). The majority of them (*n* = 16) bind to ATP1A1 subunit.

We then sought to investigate whether these drugs may enter the brain, which is the organ mainly affected in SCA1. Drugs with an 8/8 positive score in the algorithm/fingerprint combinations were predicted to cross the BBB (Appendix A). The drugs and corresponding protein targets include: Vitamin E for PP2AC, PhiKan 083 and AZD 3355 for TP53, FARNESYL for GNB1, and Bretylium and Ciclopirox for ATP1A1 (Table 2). These drugs may be used for pharmacological targeting of proteins of the central nervous system which regulate the function of the SCA1 network that is disturbed during polyQ-expanded protein aggregation.

## 4. Discussion

### 4.1. Dysregulated Pathways Associated to polyQ-Expanded ATXN1 Aggregation

Even though the disease-causing mutation in SCA1 has been previously identified, the pathogenic effects of polyQ-expanded ATXN1 are still under investigation. Our analysis indicates convergent dysregulated mechanisms in vitro and in vivo which are associated to polyQ protein aggregation in a time-dependent manner. Interestingly, our analysis indicates core pathways that are dysregulated since the beginning of protein aggregation and several signaling pathways that specifically feature the late-stage of the disease, which is characterized by irreversible disease-causing defects.

A statistically significant set of upregulated genes in both models is involved in extracellular matrix organization, one of the core dysregulated pathways which might be critical for polyQ-induced neurodegeneration. ATXN1 plays a critical role in ECM remodeling during development, affecting lung alveolarization. This suggests that polyQ-induced tissue abnormalities are not specifically restricted in the brain, but might also be present elsewhere in the body [33]. Furthermore, an increasing number of studies suggest the active involvement of ECM in neurodegeneration. ECM alterations, including the co-deposition of ECM components, may result in loss of protective perineuronal nets, increased neuronal cell death and synaptic deficiencies (reviewed in [42]).

Interestingly, late-stage polyQ aggregation affects the function of several signaling pathways (e.g., regulation of actin cytoskeleton and AGE-RAGE signaling pathway), suggesting that their dysregulation may be related to the selective loss of neuronal subtypes. This may be an indirect effect of the mutant protein or an alteration of the interaction between the polyQ-expanded protein with components/regulators of these signaling pathways, as shown for huntingtin, the polyQ-expanded protein which causes Huntington’s disease (HD) [43].

### 4.2. Network Analysis Indicates Critical Protein Nodes for SCA1 Pathogenesis

Neurodegeneration is a complex procedure involving the parallel dysregulation of several biological processes. These are accompanied by quantitative changes in protein interactions which affect the molecular architecture of biological networks inside a cell [44]. To date, a variety of techniques have been developed allowing the analysis of PPI networks [45] and can be used to study network biology of proteins involved in neurodegeneration, including ATXN1. The SCA1 network presented here is cerebellum-specific and contains network motifs, i.e., small connected sub-network patterns, at a higher frequency compared to random networks [46]. Furthermore, it is highly enriched in proteins involved in protein aggregation, validating its relevance for SCA1.

Importantly, we identified dynamic changes in the SCA1 network which are induced by the gradual aggregation of mutant ATXN1. Our analysis indicates the constant dysregulation of protein complexes involved in ion transport, accompanied by the perturbation of machineries involved in protein synthesis and oxidative phosphorylation at a middle time-point and neuronal signal transduction at a late time-point. Furthermore, the middle-stage SCA1 PPI network has the largest size, suggesting that this is a critical time-point and defects caused by polyQ aggregation beyond this stage might be irreversible. It is for this middle stage that we found the largest number of genes associated to protein aggregation in the brain.

Can we modulate disease progression by targeting critical nodes of the SCA1 network? Previous studies have shown that important nodes and potential drug targets have high degree and betweenness centralities [40,41] and their deletion is related to lethality [47]. To this end, we first ranked influential nodes of the SCA1 network based on centrality measures, as previously described [48]. Our approach indicates proteins that gradually have increased significance in the perturbed network potentially contributing to SCA1 progression and proteins that lose their significance and role in the cerebellum during mutant ATXN1 aggregation.

The proteins we selected participate in various critical processes in the nervous system and most of them are involved in the pathogenesis of HD, which is also characterized by the accumulation of protein inclusions. Rho GTPases, including CDC42 and RHOA with increased centralities in the SCA1 network, regulate neuronal cell degeneration pathways [49] and several members of these signaling pathways interact with huntingtin protein [50]. ATP1A1 with a constantly high CC value is a genetic modifier of motor deficits in HD mice [51]. Similarly, nodes with decreased centralities modulate huntingtin levels (GNB1, [52]), mediate cellular dysfunction in HD (TP53, [53]) or are involved in the translation of CAG repeat expansion mRNAs (PP2CA, [54]). Taken together, these data provide insights for selective neurodegenerative processes and suggest potential drug targets. Potentially, inhibition of nodes with increased centralities or stimulation of nodes with decreased centralities of the SCA1 network may affect the aggregation of pathogenic ATXN1.

At the moment, there is no available therapy for polyQ diseases, even though Rho/ROCK and MTOR inhibitors [55,56] or compounds entering the BBB [57] are suggested to suppress the aggregation of mutant proteins and may delay disease progression. Our approach indicates potential drug targets which are involved in the pathogenesis of HD and might be also relevant for SCA1. With regards to the computational analysis, we showed that identifying the structure and underlying motifs of the network requires several steps, most of them utilizing independently executed tools. In this context, and although efficient and reproducible, the current form of the analysis is not easily scalable, given the level of user interaction necessary to run the entire process. This evident limitation can be alleviated by ensuring interoperability between the different tools, as well as automating the execution of the pipeline. This has been partially achieved through the developed code, but full automation is planned for future iterations of the software. In conclusion, here, we propose a set of drugs that target important nodes of a cerebellum-specific SCA1 network and can also enter the BBB. These drugs could be further tested to determine whether they affect the aggregation of pathogenic ATXN1 in the brain.

## Figures and Tables

**Figure 1 genes-11-01129-f001:**
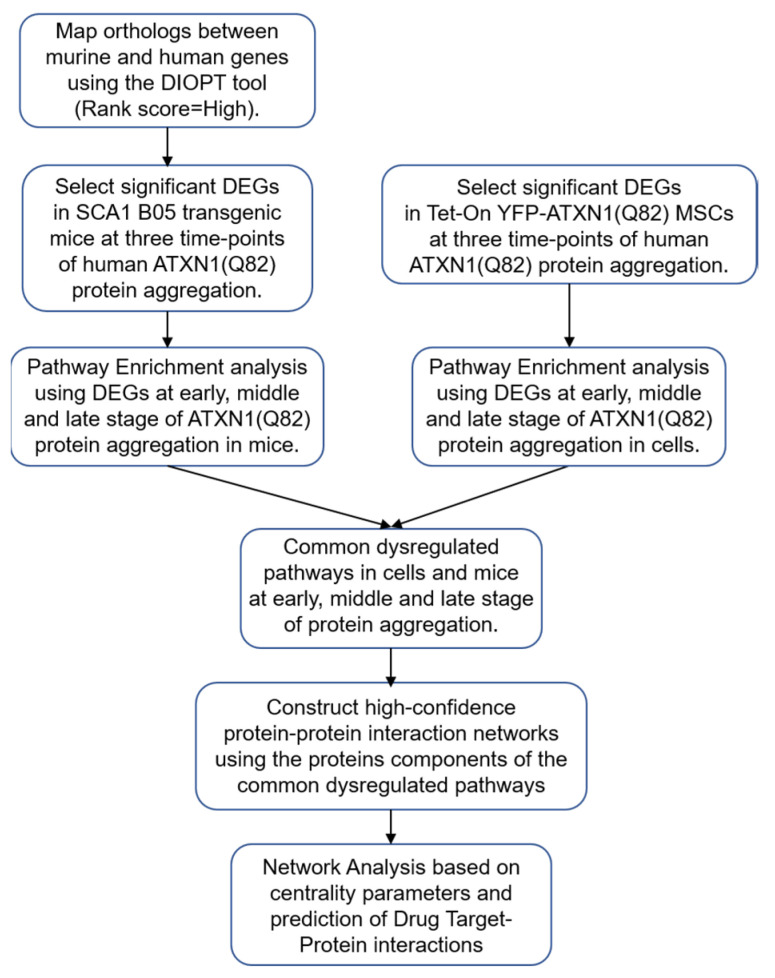
Data workflow for the construction of PPI networks associated to protein aggregation in Tet-On YFP-ATXN1(Q82) MSCs and SCA1 B05 transgenic mice.

**Figure 2 genes-11-01129-f002:**
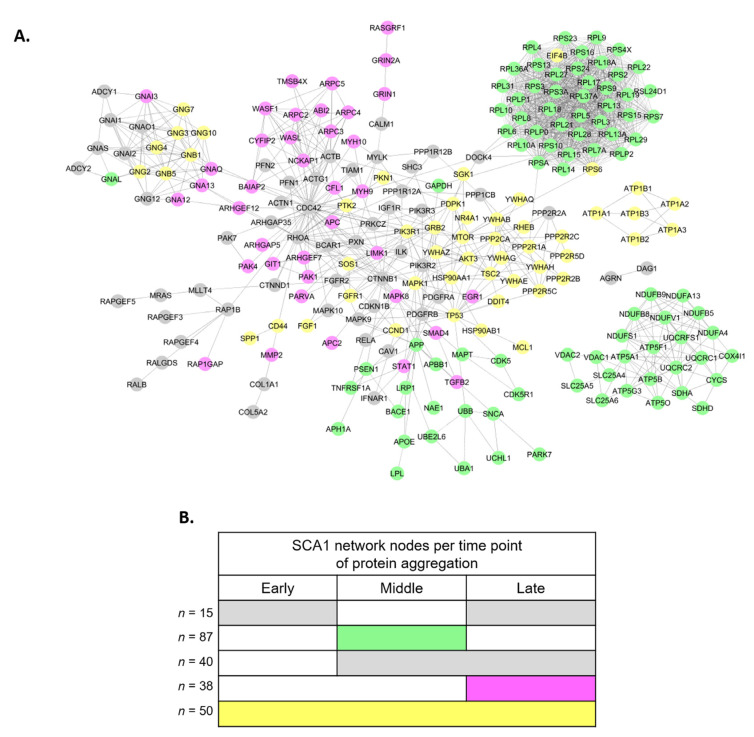
SCA1 disease PPI network. (**A**) The network contains 230 nodes (proteins) and 1432 edges (interactions). Green and magenta nodes are dysregulated at middle and late stage of protein aggregation, respectively. Yellow color indicates nodes that are dysregulated at all time-points. The network was constructed using the STRING database in Cytoscape. (**B**) Gannt chart indicating the different groups of nodes in the SCA1 disease network. Colors indicate the same groups as in (**A**).

**Figure 3 genes-11-01129-f003:**
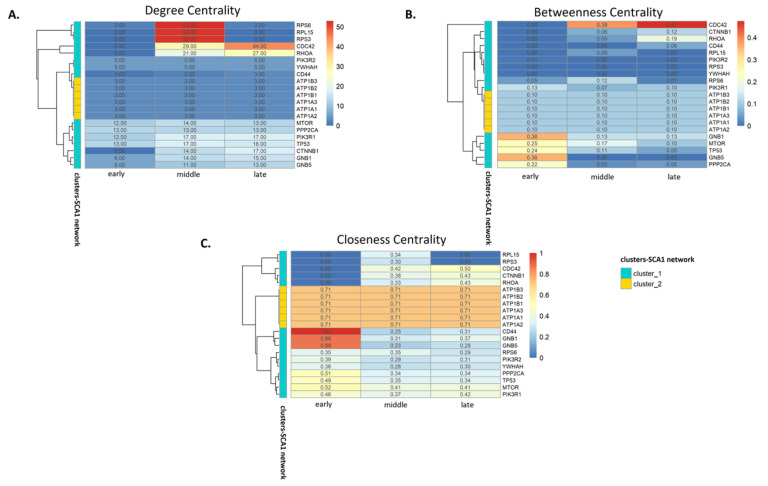
Centrality heatmaps of 21 nodes of the SCA1 disease network with the higher DC, BC and CC values per time-point. (**A**). Degree, (**B**). betweenness and (**C**). closeness centralities at early, middle or late time-point of protein aggregation. (**B**). Color range from blue (low) to red (high) indicates scaled centrality values. Chromatic code on the left side of each heatmap indicates in which cluster of the SCA1 network each node belongs.

**Table 1 genes-11-01129-t001:** Common dysregulated pathways in cells and mice at matched time-points associated with protein aggregation.

	Cells		Mice	
Enrichment Term	Overlap	*p*-Value	Overlap	*p*-Value
**A. Day 2 (D2) Cells vs. Week 5 (W5) Mice**				
Protein digestion and absorption	8/90	0.012	6/90	0.005
ECM-receptor interaction	14/82	>0.001	5/82	0.016
PI3K-Akt signaling pathway	26/341	>0.001	12/341	0.002
**B. Day 5 (D5) Cells vs. Week 12 (W12) Mice**				
Ribosome	38/137	>0.001	17/137	0.003
ECM-receptor interaction	18/82	>0.001	12/82	0.003
Focal adhesion	24/202	>0.001	21/202	0.009
PI3K-Akt signaling pathway	29/341	>0.001	30/341	0.022
Protein digestion and absorption	12/90	0.001	11/90	0.018
Alzheimer’s disease	15/168	0.002	18/168	0.011
Rap1 signaling pathway	16/211	0.001	21/211	0.015
Parkinson’s disease	11/142	0.024	14/142	0.045
**C. Day 10 (D10) Cells vs. Week 28 (W28) Mice**				
AGE-RAGE signaling pathway	9/101	0.019	11/101	0.012
ECM-receptor interaction	13/82	>0.001	9/82	0.03
Focal adhesion	21/202	>0.001	17/202	0.042
PI3K-Akt signaling pathway	26/341	0.001	27/341	0.026
Protein digestion and absorption	8/90	0.027	9/90	0.05
Rap1 signaling pathway	16/211	0.01	22/211	0.002
Regulation of actin cytoskeleton	18/214	0.002	22/214	0.002

The Table shows the common dysregulated pathways at (A) early, (B) middle and (C) late stage of protein aggregation in Tet-On YFP-ATXN1(Q82) MSCs and SCA1 B05 transgenic mice and the overlap with the components of the pathways. The analysis was performed using the Kyoto Encyclopedia of Genes and Genomes (KEGG) database.

**Table 2 genes-11-01129-t002:** Drugs that interact with components of the SCA1 protein network and are predicted to penetrate the blood brain barrier.

Target	Drug	Algorithm	Fingerprint	BBB Permeability Prediction
PPP2AC	Vitamin E	ADABoost	MACCS	BBB+
			Openbabel	BBB+
			Molprint	BBB+
			PubChem	BBB+
		SVM	MACCS	BBB+
			Openbabel	BBB+
			Molprint	BBB+
			PubChem	BBB+
TP53	PhiKan 083	ADABoost	MACCS	BBB+
			Openbabel	BBB+
			Molprint	BBB+
			PubChem	BBB+
		SVM	MACCS	BBB+
			Openbabel	BBB+
			Molprint	BBB+
			PubChem	BBB+
	AZD 3355	ADABoost	MACCS	BBB+
			Openbabel	BBB+
			Molprint	BBB+
			PubChem	BBB+
		SVM	MACCS	BBB+
			Openbabel	BBB+
			Molprint	BBB+
			PubChem	BBB+
GNB1	FARNESYL	ADABoost	MACCS	BBB+
			Openbabel	BBB+
			Molprint	BBB+
			PubChem	BBB+
		SVM	MACCS	BBB+
			Openbabel	BBB+
			Molprint	BBB+
			PubChem	BBB+
ATP1A1	Bretylium	ADABoost	MACCS	BBB+
			Openbabel	BBB+
			Molprint	BBB+
			PubChem	BBB+
		SVM	MACCS	BBB+
			Openbabel	BBB+
			Molprint	BBB+
			PubChem	BBB+
	Ciclopirox	ADABoost	MACCS	BBB+
			Openbabel	BBB+
			Molprint	BBB+
			PubChem	BBB+
		SVM	MACCS	BBB+
			Openbabel	BBB+
			Molprint	BBB+
			PubChem	BBB+

Table shows the score for each algorithm/fingerprint pairing of drugs which interact with selected nodes of the SCA1 network. Drugs with an 8/8 positive score were predicted to enter BBB.

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
