# Peer review of "Dynamics of a Protein Interaction Network Associated to the Aggregation of polyQ-Expanded Ataxin-1"

_genes, 2020, doi:10.3390/genes11101129_

Round 1
Reviewer 1 Report
Presented article is carefully prepared. The only minor points -Introduction about models of SCA1 could be more detailed and using in addition also recent literature (from 2020) and Supplementary Figure 1B looks like partially cutted off (look bottom of Figure)
Author Response
Reviewer 1
Comments and Suggestions for Authors
We thank the reviewer for his positive evaluation. Here are some minor changes to the manuscript as requested.
Presented article is carefully prepared. The only minor points:
- Introduction about models of SCA1 could be more detailed and using in addition also recent literature (from 2020).
Response: The Introduction has been modified in Lines 48-60 and 66-69. We included more information on available SCA1 models, molecular mechanisms and pathways related to SCA1 pathogenesis, as suggested by the reviewer.
- Supplementary Figure 1B looks like partially cutted off (look bottom of Figure)
Response: We would like to thank the reviewer for bringing this to our attention. We have corrected and replaced the relevant figure.
Reviewer 2 Report
In this manuscript authors tried to highlight pathways altered in 2 mammmalian models of spinocerebellar ataxia type-1. To achieve this aim they used RNAseq data at different time points already published from both models and then have carried a very comprenhensive bioinformatic analysis. In the eyes of this reviewer they have been able to extract all the relevant information from their study. This study besides its limitations is very interesting since allows to better understand the progression of the disease and will likely have a positive impact for other SCAs.
I have however, 2 major concerns
1) The first is about the statistical power of the study. I have not found the number of mice analyzed as well as the number of biological replicates in the case of the cellular model. In both cases authors only refer to other studies (ref 17 and 18). The fist one (mice model) from another group (ref 17) and the second one from an excellent work of their own using the cellular model (ref 18). It will be appropriate to add in the method section of this manuscript the n in each case. For example in reference 17, it says that 3 biological replicates per genotype and time point were used. An n of 3 in mouse studies is not very high and will suggest a relatively small statistical power. Similarly, because of the cellular model is part of their own reserach, authors might indicate the number of cells analyzed in ref 18.
2) Because of this, this reviewer thinks that some of the genes highlighted in the study will require validation using an independent approach such as qPCR or Western Blot. There is not a real validation analysis in references 17 and 18 of the RNA seq results. Only in ref 17, authors compared some of their candidates coming from the transcriptomic analysis with another models. But no validation was done in the reference 18. This is a critical aspect but I am completely opened to discuss this aspect with the authors.
Author Response
Reviewer 2
Comments and Suggestions for Authors
We would like to thank the reviewer for raising these points of criticism. Here are our responses to the comments and changes in the manuscript.
In this manuscript authors tried to highlight pathways altered in 2 mammalian models of spinocerebellar ataxia type-1. To achieve this aim they used RNAseq data at different time points already published from both models and then have carried a very comprehensive bioinformatic analysis. In the eyes of this reviewer they have been able to extract all the relevant information from their study. This study besides its limitations is very interesting since allows to better understand the progression of the disease and will likely have a positive impact for other SCAs.
I have however, 2 major concerns
1) The first is about the statistical power of the study. I have not found the number of mice analyzed as well as the number of biological replicates in the case of the cellular model. In both cases authors only refer to other studies (ref 17 and 18). The first one (mice model) from another group (ref 17) and the second one from an excellent work of their own using the cellular model (ref 18). It will be appropriate to add in the method section of this manuscript the n in each case. For example, in reference 17, it says that 3 biological replicates per genotype and time point were used. An n of 3 in mouse studies is not very high and will suggest a relatively small statistical power. Similarly, because of the cellular model is part of their own research, authors might indicate the number of cells analyzed in ref 18.
Response: We have included in Lines 78 and 86 the number of samples analyzed in each source paper (3 samples per group in SCA1 mice) and 3 samples per group in Tet-On YFP-ATXN1(Q82) MSCs. Indeed, the number of three replicates (especially in the mouse model) may seem low. However, we should take into account that all samples per group (mice or cells) show a consistent dysregulation in their transcriptome profiles due to the expression of polyQ-expanded ATXN1 and the different groups are clearly separated by PCA analysis (see relevant figures in the source papers). To support this finding, we have included the following sentence in Lines 155-156: “which show a consistent dyregulation in gene expression due to the aggregation of the mutant protein”.
2) Because of this, this reviewer thinks that some of the genes highlighted in the study will require validation using an independent approach such as qPCR or Western Blot. There is not a real validation analysis in references 17 and 18 of the RNA seq results. Only in ref 17, authors compared some of their candidates coming from the transcriptomic analysis with other models. But no validation was done in the reference 18. This is a critical aspect but I am completely open to discuss this aspect with the authors.
Response: We agree with the reviewer that independent validation of RNA-seq, especially at the protein level, is a crucial aspect, even though usually the overlap between RNA-seq and quantitative MS in the literature is approximately 30%. Concerning the mouse model, the results of the Orr lab have been indeed validated in other models of SCA1 (e.g. Friedrich et al., 2018) and a sentence with the relevant references has been added in the introduction to support their validity (Lines 60-61 and ref 18 and 21). Concerning the cell model, the results of the RNA-seq have been partially validated. The reviewer may see in Table 4 of the Redox Biology paper an independent validation of the RNA-seq data using qMS. This Table indicates that the observed down-regulation of the ribosomal genes is indeed translated into reduced protein levels, including some key nodes/ribosomal proteins of the SCA1 network presented here (e.g. RPS3 and RPS6 of Figure 3). A sentence supporting the validity of the DEGs identified in the cell model has been added in Lines 159-160. “Several of the selected DEGs, at least in the in vitro cell model, were independently validated by quantitative MS”. However, if the reviewer believes that this insertion and justification does not answer to his/her concerns, we are open to discuss the possibility to provide more experimental data for the cell model in a reasonable time frame.
Round 2
Reviewer 2 Report
Authors have addressed my concerns.